# Comprehensive Analysis of Biomass from *Chlorella sorokiniana* Cultivated with Industrial Flue Gas as the Carbon Source

**DOI:** 10.3390/molecules29143368

**Published:** 2024-07-18

**Authors:** Arjun H. Banskota, Roumiana Stefanova, Joseph P. M. Hui, Tessa Bermarija, Kevin Stemmler, Patrick J. McGinn, Stephen J. B. O’Leary

**Affiliations:** Aquatic and Crop Resource Development Research Centre, National Research Council Canada, 1411 Oxford Street, Halifax, NS B3H 3Z1, Canada; roumiana.stefanova@nrc-cnrc.gc.ca (R.S.); joseph.hui@nrc-cnrc.gc.ca (J.P.M.H.); tessa.bermarija@gmail.com (T.B.); kevin.stemmler@nrc-cnrc.gc.ca (K.S.); patrick.mcginn@nrc-cnrc.gc.ca (P.J.M.); stephen.oleary@nrc-cnrc.gc.ca (S.J.B.O.)

**Keywords:** amino acids, *Chlorella sorokiniana*, industrial flue gas, lipids, microalgae, pigments, polysaccharides, protein, sugars

## Abstract

*Chlorella sorokiniana*, isolated from a pond adjacent to a cement plant, was cultured using flue gas collected directly from kiln emissions using 20 L and 25000 L photobioreactors. Lipids, proteins, and polysaccharides were analyzed to understand their overall composition for potential applications. The lipid content ranged from 17.97% to 21.54% of the dry biomass, with carotenoid concentrations between 8.4 and 9.2 mg/g. Lutein accounted for 55% of the total carotenoids. LC/MS analysis led to the identification of 71 intact triacylglycerols, 8 lysophosphatidylcholines, 10 phosphatidylcholines, 9 monogalactosyldiacylglycerols, 12 digalactosyldiacylglycerols, and 1 sulfoquinovosyl diacylglycerol. Palmitic acid, oleic acid, linoleic acid, and α-linolenic acid were the main fatty acids. Polyunsaturated fatty acid covers ≥ 56% of total fatty acids. Protein isolates and polysaccharides were also extracted. Protein purity was determined to be ≥75% by amino acid analysis, with all essential amino acids present. Monomer analysis of polysaccharides suggested that they are composed of mainly D-(+)-mannose, D-(+)-galactose, and D-(+)-glucose. The results demonstrate that there is no adverse effect on the metabolite profile of *C. sorokiniana* biomass cultured using flue gas as the primary carbon source, revealing the possibility of utilizing such algal biomass in industrial applications such as animal feed, sources of cosmeceuticals, and as biofuel.

## 1. Introduction

Carbon dioxide (CO_2_) is a greenhouse gas (GHG), and its excess emission, either due to natural processes or human activities, plays an important role in contributing to global warming [1]. Various strategies have been implemented at the national and global level to mitigate climate change and reduce carbon emissions in the atmosphere, such as by improving energy efficiency technologies, using low-carbon fuel, or deploying renewable fuel. Technologies for capturing atmospheric CO_2_ or utilizing flue gas from the emission source are also in progress. Microalgae are being considered for use as a biological tool to capture industrial CO_2_ because of their ability to sequester inorganic carbon (CO_2_) through photosynthesis [2]. These microscopic organisms are also capable of producing high-value products with possible uses in food, feed, and fuel applications [3].

*Chlorella* is a well-studied microalgae genus that has garnered considerable attention both in laboratory research and in industrial-scale cultivation because of the rapid growth rate and high degree of adaptability to different environmental conditions demonstrated by various *Chlorella* species [4]. They are a rich source of proteins, omega-3 fatty acids, vitamins and other nutrients making them a valuable source for various applications, including animal feed, human consumption and nutraceuticals [5,6]. *C. sorokinana* is among the species of *Chlorella* investigated for producing protein for aquafeed and biofuel purposes [7,8]. It has been reported to have high photosynthetic productivity, i.e., 34.4 g of dry biomass/m^2^ of installation area/day in batch culture when cultivated in a helical tubular photobioreactor [9]. *C. sorokinana* can grow both in heterotrophic and mixotrophic conditions and shows promise in wastewater treatment applications with high removal rates of nitrogen and phosphorous [10]. The biomass of *C. sorokiniana* contains a relatively high lutein content as compared to other carotenoids and random mutagenesis has led to the isolation of high lutein-yielding mutants of *C. sorokiniana* with potential for commercial use [11]. Banskota et al. also reported bioactive glycolipids from the *C. sorokiniana* UTEX 1230 strain, especially monogalactosylmonoacylglycerols (MGMGs) with anti-inflammatory properties and monogalactosyldiacylglycerols (MGDGs) with pancreatic lipase-inhibition properties [12,13].

A number of microalgal strains have been studied for capturing CO_2_ from industrial flue gas including *Arthrospira* (*Spirulina*) sp. *Chlamydomonas renhardtii*, *Chlorella fusca*, *Chlorella minutissima*, *Chlorella pyrenoidosa*, *Chlorella sorokininana*, *Chlorella vulgaris*, *Galdieria sulphuraria*, *Haemtococcus pluvialis*, *Nannochloropsis oculatata*, *Scenedesmus quadriculata,* and *Tetraselmis suecica* [14,15,16]. Laboratory studies have shown that CO_2_ sequestration/capture efficiency/efficacy can be affected by many factors including temperature, pH and concentrations of CO_2_, SOx, and NOx in the input gases [14], all of which can vary greatly in industrial flue gases [14]. The CO_2_ biofixation rate from flue gases by those microalgal species range from 72 mg L^−1^ d^−1^ to 435 mg L^−1^ d^−1^ [16]. Chauhan et al., screened 13 microalgal strains to evaluate their gradual acclimation capacity to toxic flue gas and among the tested algal strain, *Micractinium pusillum* KMC8 and *Scenedesmus acutus* NCIM5584 were found to accumulate nitrite as fixed nitrogen. The KMC8 strain showed a high tolerance to flue gas, a higher biomass yield (1.32 g L^−1^), and high neutral lipid accumulation [17]. Similarly, Nagappan et al. (2020) thoroughly described the composition of cement kiln gas and its effect on microalgal strain tolerance [18]. Wang et al. (2025) and Ma et al. (2022) described the latest development of CO_2_ fixation by microalgae and the mechanisms and pathways of CO_2_ sequestration [19,20]. To improve the productivity of the biomass for biofuel production or CO_2_ fixation, large-scale outdoor cultivation of microalgae has been performed using Spirulina and four other microalgal strains [21,22].

In addition, Mortensen et al. (2016) cultured *C. sorokiniana* under flue gas at two different concentrations, achieving equal biomass production after 4 days of growth as compared to the control culture with 5% pure CO_2_ gas [23]. Moreover, Kumar et al. (2014) observed that high CO_2_ and H_2_S concentration in flue gas had an inhibitory effect on the growth of *C. sorokiniana.* To overcome this problem, they used diluted flue gas [24]. Lizzul et al. (2014) further explored exhaust gases and wastewater to replace conventional feedstock to produce biomass from *C. sorokiniana* [25]. Dickinson et al. (2023) also studied 28 microalgae strains isolated from the area surrounding a commercial cement plant. After down selecting growth rate and flue gas tolerance, *C. sorokiniana* (strain SMC-14M) appeared to be the most productive in the presence of kiln gas emissions collected from a cement plant; thus, this strain was selected for further biomass production to study its chemical composition [26]. Most previous studies with flue gases primarily focus either on biomass production or lipid analysis for biofuel application even though the biomass has other possible applications such as animal feed, aquafeed and nutraceuticals. No comprehensive study has been performed on the chemical composition of *Chlorella* spp. biomass derived from flue gas as a carbon source. In the present study, we explore the in-depth chemical characterization of *C. sorokiniana* biomass cultured into two types of photobioreactors with flue gas directly collected from cement kiln emissions. The high-value products such as pigments including carotenoids, both polar and non-polar lipids were characterized by proton nuclear magnetic resonance (^1^H NMR), high-performance liquid chromatography (HPLC), liquid chromatography/mass spectrometry (LC/MS) and gas chromatography (GC) analyses. Protein isolates (PI) and polysaccharides were also extracted and analyzed.

## 2. Results and Discussion

*C. sorokiniana* (SMC-14M) was isolated from a water sample collected from ponds near the St. Mary’s Cement Plant, ON, Canada, and phylogenetic analysis was conducted by sequencing a short fragment of DNA for the identification of the microalga [26]. At the cement plant, a series of 20 L photobioreactors and a 25,000 L pilot-scale photobioreactor were used to cultivate SMC 14M biomass using kiln flue gas as the carbon source. *C. sorokiniana* was found to be among the best algal strains for growth rate and flue gas tolerance among the previously tested microalgae [26] and was therefore selected for scaleup biomass production in the 25,000 L pilot scale photobioreactor and characterization of algal biomass for possible industrial application. Although *C. sorokiniana* (SMC-14M) was not cultured using control CO_2_ for comparison, its biomass production and growth rates were ~0.9 g L^−1^ and 1.18 d^−1^, respectively, when cultured using flue gas [26]. Biomass samples collected from both size photobioreactors were dark green in color (Appendix A) most likely due to chlorophyll. Concentrations of chlorophyll a in the algal biomass collected from 20 L and 25,000 L photobioreactors were 33.4 mg/g and 18.1 mg/g of dry biomass, respectively. Chlorophyll b concentrations were between 7.6 and 8.8 mg/g (Figure 1). High concentrations of chlorophyll a in *C. sorokiniana* biomasses cultured using flue gas were in alignment with the reported data by Negi et al. (2016) when they studied the impact of nitrogen limitation on photosynthesis and lipid accumulation in *C. sorokiniana* [27]. Lutein, α-carotene, and β-carotene were the main carotenoids detected in the tested algal biomass. The total carotenoid concentrations were 9.2 mg/g and 8.4 mg/g in the biomass derived from 20 L and 25,000 L photobioreactors, respectively. Lutein alone covers more than 55% of the total carotenoid, i.e., ≥5.1 mg/g of the dry biomass collected from both photobioreactors. The lutein concentrations in the tested biomasses are in good agreement with high lutein-producing *C. sorokiniana* mutants reported in the literature, which produce 7.0 mg/g of dry biomass [11]. The results suggest that *C. sorokiniana* cultured using flue gas may be a potential source of lutein and other carotenoids reported to have health benefits [28]. The moisture and ash content of both biomasses were also measured; the moisture content of samples from the 20 L and 25,000 L photobioreactors were 1.4 ± 0.4 and 3.7 ± 0.3, respectively. The ash contents were 6.4 ± 0.8 and 5.7 ± 0.3%, respectively.

The algal biomass collected from the 20 L photobioreactors showed a higher percentage of lipid content (21.5 ± 1.5%) than biomass collected from the 25,000 L photobioreactor (18.0 ± 0.4%). Though the two biomasses have slightly different lipid contents, the lipid percentages are in a similar range as reported in other studies of *C. sorokiniana* cultured using flue gas (21.1%) or even in heterotrophic culture (10–25%) [24,29]. A slightly higher lipid content (29.5%) was also reported by Qiu et al. (2017) when they studied the effect of pH on lipid production in *C. sorokiniana* [30]. The difference in lipid content we observed compared to those found in other studies may be due to the strain itself or effects caused by other external factors such as culture type, media, nutrient supply, light intensity, pH, etc. [31]. The fatty acid composition of the lipids extracted from both biomasses was analyzed by gas chromatography-flame ionization detector (GC-FID) after transesterification. Palmitic acid (16:0), oleic acid (18:1), linoleic acid (18:2), and α-linolenic acid (18:3) were the main fatty acids identified accounting for 88% of the total fatty acids (Figure 2), and a similar composition has been reported in two strains of *C. sorokiniana* (KNUA122 and KNUA114) cultured at various temperatures [32]. Higher linoleic acid (44.8%) was observed in lipids extracted from biomass collected from 20 L photobioreactors as compared to only 29.6% linoleic acid from the 25,000 L photobioreactor. The compensatory α-linolenic acid concentration was higher in the biomass collected from the 25,000 L photobioreactor (Figure 2). In both algal biomasses, the polyunsaturated fatty acids (PUFA) concentration was the highest followed by saturated fatty acids (SFA) and monounsaturated fatty acids (MUFA) (Figure 3). The absolute values of individual fatty acids (mg/g of the lipid fraction) are listed in Appendix A.

The lipid extracted from both biomasses was further partitioned into neutral lipid, glycolipid and phospholipid fractions by silica gel-based solid phase extraction (SPE) as described by Ryckebosch et al. (2012) to characterize exact lipid composition [33]. The lipid fraction clearly demonstrated that both biomasses have higher neutral lipid (Figure 4). Glycolipid was slightly higher (26.0%) in biomass grown in the 25,000 L photobioreactor as compared to 20 L photobioreactor (23.4% of the total lipid). The phospholipid, on the other hand, was higher in biomass from the 20 L (31.4%) reactors compared to the 25,000 L (23.6%) photobioreactor. The ^1^H NMR spectra of lipid fractions are shown in Figure 5 (20 L photobioreactor) and in Appendix A (25,000 L photobioreactor). Major signals in the ^1^H NMR spectra of neutral lipid fractions at 0.75–3.00 and 5.40 ppm suggest the presence of fatty acid acyl signals. Methine signals of glycerol backbone at 5.23 ppm indicated the neutral lipid contained triacylglycerols (TAGs) [34]. The glycolipid fraction eluted with acetone showed a signal corresponding to chlorophyll or pheophytin-a in its ^1^H NMR spectrum [35]. The proton NMR signals of the phospholipid fraction indicated the presence of sugar signals between 3.50 and 4.50 ppm, and fatty acid acyl signals (at 0.75–3.00 and 5.40 ppm) suggested that the fraction contains mainly polar lipids.

Ultra-high-performance liquid chromatography/high-resolution mass spectrometry (UHPLC/HRMS) with electrospray ionization (ESI) was used to characterize individual lipids present within the neutral lipid, glycolipid, and phospholipid fractions. As suggested by the ^1^H NMR spectra, the neutral lipid fraction was analyzed for triacylglycerols (TAGs). The total ion current (TIC) chromatograms of neutral lipids are shown in Appendix A, and the major ion peaks eluted between 1.00 and 1.25 min belong to pheophytin-a with *m*/*z* 871.5733 [36]. TAGs were eluted between 1.5 and 4.0 min. The ammonium adduct ions were selected for the identification of TAGs using the LIPID MAPS database [37], which produces prominent fragment ions leading to the structure elucidation of TAGs. A list of TAGs with observed [M + NH_4_]^+^ ions matched with the database is presented in Table 1. In total, 71 TAGs were detected and their fatty acid acyl side chains were identified by analyzing fragmentation ions as described previously [34]. For example, as shown in Appendix A, a TAG with molecular ion peak at *m*/*z* 848.7705 [matched with TAG (50:2) with *m*/*z* 848.7702 in LIPID MAPS database] eluted at 3.10 min had the fragmentation ions at *m*/*z* 551.5029 (neutral loss of 18:2) and 575.5071 (neutral loss of 16:0). The neutral loss of 18:2 and 16:0 suggested the acyl side chain of TAG (50:2) must be 18:2/16:0/16:0. Moreover, based on the peak area of the ammonium adduct ions, the relative abundance of individual TAGs was also measured, which suggested that TAGs with fatty acid acyl chain 18:2/18:0/16:0; 18:2/16:0/16:0 were in high abundance. To the best of our knowledge, this is the first report of the characterization of TAGs in *C. sorokiniana*.

The UHPLC/HRMS analysis of the glycolipid fraction of both biomasses led to the identification of 9 monogalactosyldiacylglycerols (MGDGs), 12 digalactosyldiacylglycerols (DGDGs) and 1 sulfoquinovosyl diacylglycerol (SQDG). Free fatty acids were also detected in the glycolipid fraction in addition to strong signals of chlorophyll degradation products eluted at 3.50–4.50 min as shown in TIC (Appendix A). Ammonium adduct ions were observed for all MGDGs, DGDGs, and SQDG and were searched in the LIPID MAPS database [37]. Analyzing fragmentation ions of both molecular ions, i.e., [M + NH4]^+^ and [M + Na]^+^ led to the identification of fatty acid acyl side chains and their position. For example, as shown in Figure 6, the molecular ion peak at *m*/*z* 930.6160 eluted at 6.25 min matched with the LIPID MAPS database corresponding to the [M + NH_4_]^+^ ion of DGDG (34:4) with *m*/*z* 930.6148 [30]. The fragment ion at *m*/*z* 571.4764 represents the loss of NH_3_ followed by the loss of the two galactosyl units. The fragmentation ions at *m*/*z* 309.2436 and 337.2559 were formed by losses of fatty acids 16:2 and 18:2, respectively, from the [M + H − Gal − Gal]^+^ ion, which were diagnostic peaks to identify the fatty acid acyl side chain (Figure 6). Moreover, the sodium adduct ion at *m*/*z* 935.5710 provides fatty acid neutral loss fragments at *m*/*z* 683.3603 and 655.3301 corresponding to the loss of fatty acids 16:2 and 18:2, respectively (Figure 7). Based on the greater intensity of the *m*/*z* 655.3301 ion (18:2), the DGDG has the acyl group 18:2 in position *sn*-1 and 16:2 in position *sn*-2, according to previous reports [13,38,39]. Accordingly, the regiospecific positions of the fatty acid acyl group in the glycerol backbone of DGDGs and MGDGs were identified. Based on the intensities of individual [M + NH_4_]^+^ ions, the relative abundance of MGDGs and DGDGs were also measured and are listed in Table 2. The MGDGs and DGDGs were evenly distributed in the glycolipid fractions of biomass produced in the 25,000 L photobioreactor, i.e., 51.2% and 48.8%, respectively. The concentration of DGDGs (59.4%) is slightly higher in algal biomass cultured in 20 L photobioreactors as compared to MGDGs (40.6%). The major MGDG was 34:4 which covers > 14.5% of the total galactolipids. In a previous study, Banskota et al. reported 12 MGDGs from *C. sorokiniana* UTEX 1230 cultured in f/2 media [13]. In the current study, 10 out of 12 MGDGs plus MGDG (34:6) were detected in the phospholipid fraction, suggesting no impact on the galactolipids profiles when flue gas is used to culture *C. sorokiniana*. A study by Widzgowski et al. (2020) demonstrated that the composition of these membrane lipids directly correlates with the light intensities that cells are exposed to during culture [40]. Isotopic labeling experiments revealed that different light treatments in *C. sorokiniana* caused shifts within the glycolipid composition, and these shifts are species-specific. The slight difference in the individual glycolipid abundance in two reactor sizes may be due to the difference in light used in the 20 L and 25,000 L photobioreactors. Even though the presence of MGDG, DGDG, phosphatidylethanolamine (PE), phosphatidylcholine (PC), and phosphatidylglycerol (PG) was reported in *C. sorokinina* either by Fourier-transform ion cyclotron resonance mass spectrometry (FT-ICR-MS) or thin-layer chromatography (TLC), no characterizations of these molecules have been conducted in previous studies [31,40,41] except MGDG [13]. It is worth mentioning here that this is the first report of characterization of DGDGs from *C. sorokinina*.

Galactolipids are parts of membrane lipids reported to have a wide range of biological properties, including anti-tumor-promoting, anti-inflammatory, anti-bacterial, anti-obesity, and anti-microfouling properties [13,42]. Junpeng et al. (2020) used heterotrophic cultivation to produce MGDG for value-added product development and were able to produce 16.5 mg/L/day MGDG in *C. sorokiniana* [41]. We further studied the lipids extracted with chloroform/MeOH of algal biomass collected from a 25,000 L photobioreactor by conventional silica gel-based column chromatography separation. The major components in each fraction were identified either by HPLC or LC/MS analysis with a single quad MS detector (Appendix A). The results clearly demonstrated that *C. sorokiniana* cultured with flue gas has >40% polar lipids, mainly glycolipids, that could be used for commercial applications.

The phospholipid fractions are comprised of lyso-phosphatidylcholines (LPCs), phosphatidylcholines (PCs), MGDGs, DGDGs and SQDG. In total, 8 LPCs, 10 PCs, 1 MGDG, 12 DGDGs, and 1 SQDG were identified (Table 3). All digalactosyldiacylglycerols (DGDGs) identified in the phospholipid fraction were also present in the glycolipid fraction. Monogalactosyldiacylglycerol (MGDG 34:6) is the only MGDG detected in the phospholipid fraction. The presence of galactolipids in both glycolipid and phospholipid fractions suggested the high abundance of galactolipids in *C. sorokiniana*. LPCs were eluted between 2.50 and 4.50 min, while PCs were eluted after 5.20 min, as shown in corresponding TICs (Appendix A). The identifications of LPCs and PCs were made based on the strong diagnostic fragment at *m*/*z* 184.0734 in positive mode which belongs to the protonated phosphocholine head group [43,44]. The individual LPCs and PCs were further confirmed based on the accurate masses of the molecular adduct ions [M + H]^+^ that match with the LIPID MAPS database [37]. The *sn*-1 position of the FA acyl chain was confirmed based on the significantly higher intensity peak of ions at *m*/*z* 104.1078 as compared to peak ion at *m*/*z* 125.0005 (Appendix A) because it has been reported that over 30-fold differences in the peak intensity ratio of product ions at *m*/*z* 104 and 147 corresponds to a sodiated *sn*-1-acyllysophatidycholine in comparison to *sn*-2-acyllysophatidycholine [45]. PCs were also identified based on the diagnostic fragment at *m*/*z* 184.0734 in a positive mode (Appendix A) and matched with the LIPID MAPS database [30]. However, we were unable to identify the position of the FA acyl chain in PCs due to the poor fragmentation observed for [M + COO]^–^ ions in negative mode (Appendix A). Phospholipids possess anti-oxidant activity and are part of our food system and health regarding memory, improving immunity and preventing cardiovascular disease [46]. *C. sorokiniana* biomass may be a possible source of phospholipids for animal feed applications and has already been studied for tilapia feeds because of its high protein and polyunsaturated fatty acid (PUFA) content [47].

The residual biomass recovered after lipid extraction by chloroform/MeOH was further used for protein and polysaccharide extraction. Protein isolates (PIs) were extracted by treatment with 1M NaOH followed by isoelectric precipitation. In total, 13.2% PI was extracted from the residual algal biomass collected from the 25,000 L photobioreactor. The supernatant after PI precipitation was further diluted with EtOH, which gave 3.5% crude polysaccharides. The percentage of PI and crude polysaccharide from biomass cultivated in the 20 L photobioreactors were 8.6% and 4.3%, respectively. PIs are light green in color, whereas crude polysaccharides extracted from materials from both 20 L and 25,000 L photobioreactors are colorless, as shown in Figure 8. The amino acid analysis results suggested that the purity levels of the PI isolated from biomass from the 20 L and 25,000 L photobioreactors were 75.0% and 85.2%, respectively. The quantitative data of individual amino acids in PIs are shown in Table 4 and the HPLC chromatograms are shown in Appendix A. Fourier-transform infrared spectroscopy (FT-IR) spectra of PIs isolated from *C. sorokiniana* were identical, showing two strong absorption peaks around 1630 cm^−1^ and 1520 cm^−1^ belonging to amide I and II (Appendix A), suggesting their similarity. Aspartic acid (Asp), glutamic acid (Glu), glycine (Gly), Arginine (Arg), valine (Val), leucine (Leu), and phenylalanine (Phe) were the major amino acids with concentrations of > 50 mg/g of crude protein identified in PIs isolated from *C. sorokiniana* cultured using flue gas (Table 4). The production of higher Gly, Ala, and Leu was observed by Ballesteros-Torres et al. (2019) when *C. sorokiniana* was cultured using wastewater [48]. Moreover, all essential amino acids were found in PIs except tryptophan, which was not a part of the analysis. It is worth mentioning here that an earlier study of the same strain cultured in a 1000 L photobioreactor, which directly tested algal biomass samples for their amino acid content, had similar results, yielding a crude protein content of 51–52% as determined by nitrogen count [26]. Cao et al. also studied protein and polysaccharide optimization in *C. sorokiniana* using the Bradford method for protein quantification observing 12.8% protein (biomass 3.58 g/L and protein yield 0.46 g/L) in optimized biomass [49]. To the best of our knowledge, this is the first report of PI extraction and characterization from *C. sorokiniana*.

The percentage of crude polysaccharide was significantly lower (only 3.5% of residual biomass of 25,000 L photobioreactor) as compared to the literature data (35%) when the sequential subcritical hydrothermal liquefaction (SEQHTL) extraction technique was applied [50]. The FT-IR signals suggested that crude polysaccharides extracted/isolated from the biomasses collected from 20 L and 25,000 L photobioreactors were almost identical, having typical O-H stretching and C-H stretching signals at 3500 cm^−1^ and 2900 cm^−1^, respectively, corresponding to the OH and CH groups (Appendix A). More than 90% of the polysaccharides extracted from *C. sorokiniana* by the SEQHTL process were characterized as 1→4 linked glucan [51]. The crude polysaccharides extracted from the residual biomasses were also analyzed for their monomer composition by gas chromatography flame ionization detector (GC-FID). The polysaccharide extracted from *C. sorokinina* cultured in 20 L photobioreactors was mainly composed of D-(+)-galactose, D-(+)-glucose and D-(−)-fructose; their relative intensities were 55.1, 23.6 and 7.0%, respectively (Figure 9). The relative percentages of D-(+)-galactose, D-(+)-glucose and D-(−)-fructose were 41.3, 13.5 and 7.5%, respectively, in polysaccharide extracted from algal biomass cultured in the 25,000 L photobioreactor. Even though major monomers were identified, the structure of polysaccharides extracted in the current study at room temperature remains unknown and further study is needed for their full characterization.

## 3. Materials and Methods

### 3.1. General

The ^1^H NMR spectra were measured on a Bruker 700 MHz spectrometer using a 5 mm cryogenically cooled probe. HPLC and LC/MS analyses were carried out on an Agilent 1200 Series system equipped with a diode array and single quad mass detector. Gas chromatography (GC) analyses were carried out on an Agilent Technologies 7890A GC spectrometer with a FID detector (Agilent Technologies Inc., Santa Clara, CA, USA). FT-IR data were recorded using a Nicolet iS10 FTIR (Thermo Fisher Scientific, Waltham, MA, USA). HPLC-grade chloroform, methanol, acetonitrile, acetone and MilliQ water were used for extraction and analytical work.

### 3.2. Algal Culture

The freshwater chlorophytic alga *C. sorokiniana* was cultured in two different photobioreactor (PBR) systems at the St. Mary’s cement plant in St. Mary’s, Ontario. Eighteen ‘bucket’ PBRs were purpose-built and consisted of a 20 L plastic bucket fitted at the top with an air-cooled, monochromatic LED (peak emission 640 nm) as a light source for photosynthesis. The 20 L bucket PBRs were used for strain screening and therefore directly aerated with kiln gas containing ~5% CO_2_. Ten liters of sterilized water was added to each PBR followed by amendment with f/2 growth medium and inoculation with 0.25 L flask cultures of *C. sorokiniana*. At the conclusion of the screening trial, the cultures were pooled and the biomass harvested through a rotary separator. After collection, the biomass was frozen at −20 °C until analysis. A pilot-scale PBR (total volume 25,000 L) consisting of two 12,500 L compartments separated by a central bulkhead was used in a series of growth trials using kiln gas to test the up-scalability of algal carbon capture and conversion. This PBR was fitted with externally-mounted monochromatic LEDs (peak emission 640 nm) cooled with chilled glycerol. Six thousand liters of sterilized water was added to one side of the PBR which was amended with f/2 growth medium and inoculated with a culture of *C. sorokiniana* grown in a separate 1000 L PBR in the same medium and aerated with kiln gas. A full complement of f/2 nutrients, excluding trace metals and vitamins, was added to the culture every second day. The pH varied around an average value of 7.2 over the eight-day cultivation period. At the end of this trial, 1200 L of culture was harvested through a rotary separator and the collected biomass was frozen at −20 °C until analysis.

### 3.3. Moisture, Ash, Lipid Content and Lipid Class Separation

Approximately 100 mg freeze-dried algal biomass was placed in a porcelain crucible and heated at 110 °C for 12 h in triplicate. The dried sample was transferred into a desiccator, and the weight was measured after cooling down to room temperature to calculate moisture content using the formula describe below (1). The resulting algal biomass was further heated at 550 °C for 12 h in a Muffle furnace. The remaining ash was slowly cooled down in a desiccator to room temperature and the weight was measured gravimetrically to calculate ash content using the formula describe below (2). Lipid content was measured by the Folch method with modifications [52]. In brief, the freeze-dried algal biomass (~100 mg) was extracted at room temperature homogenizing with CHCl_3_/MeOH (2:1, 1 mL × 3) using a bead beater (Bead Mill_24_, Fisher Scientific, Hampton, NH, USA) in a 2 mL Lysing matrix Y tubes (3 × 1 min cycles) in triplicate. The combined lipid extract was dried under nitrogen and total lipid content was calculated using the formula described below (3). The lipid was further separated into three fractions, i.e., neutral lipid, glycolipid and phospholipids using silica gel solid-phase extraction (SPE) as described previously, eluting with chloroform, acetone and MeOH [34].
Moisture content (%) = (sample weight − dry sample weight)/sample weight × 100(1)
Ash content (%) = Ash weight/Sample weight × 100(2)
Lipid content (%) = CHCl_3_/MeOH extract weight/Sample weight × 100(3)

### 3.4. Ultra-High-Performance Liquid Chromatography/High-Resolution Mass Spectrometry (UHPLC/HRMS) Analysis

UPLC-HRMS data were acquired on an UltiMate 3000 LC system coupled to a Q-Exactive^TM^ hybrid Quadrupole Orbitrap Mass Spectrometer (Thermo Fisher Scientific, Waltham, MA, USA) equipped with a HESI-II probe for electrospray ionization (ESI). A Thermo Hypersil Gold C8 column (100 × 2.1 mm, 1.9 μm) was used for TAG separation at 40 °C with an acetonitrile/isopropyl alcohol (IPA) gradient as described previously [34]. The glycolipids and phospholipids were separated with 10 mM ammonium acetate pH 5.0/methanol gradient while 5 mM ammonium formate in IPA/de-ionized water/methanol 1/2/7 (*v*/*v*) was delivered constantly at 100 μL/min to the MS via a metering pump as described earlier [53]. The MS condition is identical as described earlier [34,53].

### 3.5. Fatty Acid Analysis

Fatty acid analysis was carried out according to the AOAC official method 991.39 with modifications [54]. In brief, lipid from algal biomass was dissolved in methanol containing internal standard (IS) methyl tricosanoate, then trans-esterified by treating with 1.5 M NaOH solution followed by 14% BF_3_ solution in methanol. Three independent experiments were performed. Fatty acid methyl esters (FAMEs) were extracted with hexane and used for GC analysis. Fatty acid content in the lipid was calculated and expressed as μg/mg lipid using the formula described below (4).
Fatty acid (mg/g) = (A_X_ × W_IS_ × CF_X_/A_IS_ × W_S_ × 1.04) × 1000(4)
where A_X_ = area counts of fatty acid methyl ester; A_IS_ = area counts of internal standard; CF_X_ = theoretical detector correlation factor is 1 except for EPA or DHA (0.99 for EPA, 0.97 for DHA); W_IS_ = weight of IS added to sample in mg; W_S_ = sample mass in mg; and 1.04 is factor necessary to express the result as mg fatty acid/g sample.

### 3.6. Pigment Analysis

Approximately 10 mg of freeze-dried sample was extracted at room temperature by homogenizing with CHCl_3_/MeOH (1:1, 1 mL × 3) using a bead beater (Bead Mill_24_, Fisher Scientific) in a 2 mL Lysing matrix Y tubes (3 × 1 min cycles). The combined extract was dried under nitrogen and re-dissolved in MeOH (1.0 mL) for HPLC analysis. Carotenoids and chlorophyll analysis was performed using a YMC Carotenoid column (5 μm, 2 × 250 mm, 181 YMC Co., Ltd., Kyoto, Japan) eluting with 50 mM NH_4_OAc in MeOH/tertiary butyl methyl ether (TBME) linear gradient 5 to 65% B in 30 min at 0.2 mL/min flow rate for 60 min. Standard curves of chlorophyll a, chlorophyll b, lutein, α-carotene and β-carotene purchased from ChromaDex (Longmont, CO, USA) were used for quantification, and absorbance at 450 nm and 720 nm were used for carotenoid and chlorophyll analysis, respectively. The concentration of unknown carotenoids was calculated based on the response for lutein.

### 3.7. Protein Isolate (PI) and Polysaccharide Extraction

Freeze-dried algal biomass (15.0 g) was extracted with chloroform/MeOH (1:1, 100 mL × 3) by sonicating (15 min) at room temperature. The chloroform/MeOH extract was subjected to purification and the residual biomass was further used for protein and polysaccharide extraction. In brief, residual biomass (10 g) was mixed with 1M NaOH (100 mL) and stirred overnight at room temperature. The mixture was centrifuged at 2844× *g* for 20 min, the supernatant was collected, and the pH was adjusted to around 2.0 by adding 1M HCl followed by TFA. The resulting mixture was kept in the fridge overnight and protein isolate (precipitate) was collected by centrifugation at 2844× *g* for 20 min. The protein isolate was washed with Milli-Q water twice and freeze-dried. The protein isolate yield was 1.32 g (13.2% of the residual biomass). After protein separation, the supernatant was diluted 7:1 with EtOH to precipitate polysaccharides, which were separated by centrifugation at 2844× *g* for 20 min, washed with Milli-Q water twice, and freeze-dried. The yield of polysaccharides was 0.35 g (3.5% of residual biomass 25,000 L photobioreactor). The percentage of PI and crude polysaccharide from biomass cultivated in the 20 L photobioreactors were 8.6% and 4.3%, respectively.

### 3.8. Amino Acid (AA) Analysis

Amino acid analysis of the crude protein was performed according to Waters AccQ.Tag method [55]. In brief, protein isolate (~20 mg) containing internal standard nor-leucine (Nleu) and a few crystals of phenol was hydrolyzed with 6N boiling HCl (1.0 mL) at 120 °C for 24 h. The reaction was cooled down to room temperature, filtered through cotton and diluted to 10 mL using Milli-Q water. The hydrolyzed solution (20 μL) was dried using Genovac. The dried hydrolyzed sample was derivatized according to the Waters AccQ-Tag protocol as follows: the sample was dissolved in 80 µL of borate buffer (0.2 M, pH 8.8) and 20 µL of Waters AccQ Fluor reagent was added. The mixture was vortexed and heated at 55 °C for 10 min and subjected to HPLC analysis on an Agilent 1260 LC system with fluorescence detector. Waters amino acid hydrolysate standard was derivatized in the same way and used as an external standard for quantification. The amino acid separation was carried out using a Waters AccQ-Tag C18 column (4 µm, 3.9 × 150 mm). The solvent system consisted of Eluent A: aqueous buffer and Eluent B: acetonitrile/water (3:1). The flow rate was 0.7 mL/min and the column temperature was maintained at 55 °C. The injection volume was 1 µL. The following solvent gradient was used: 0–0.5 min, 98% A–2.0% B; 15.0 min, 93% A–7.0% B; 19.0 min, 90% A–10% B; 32.0 min, 67% A–33% B; 33.0 min, 67% A–33% B; 34.0 min, 100% B; 37 min, 100% B; 38 min, 100% A, 64 min, 100% A. Agilent OpenLab CDS ChemStation Edition (Rev.C.01.07.SR3) software was used for data acquisition and analysis.

### 3.9. Sugar Analysis

Sugar analysis of polysaccharides extracted from the algal biomass was performed by GC after hydrolysis of polysaccharides followed by acetylation. In brief, polysaccharides (~20 mg) were hydrolyzed with 2M trifluoroacetic acid (TFA, 1.5 mL) at 100 °C for 1 h. The hydrolyzed product was dried under nitrogen gas and dissolved in Milli-Q water (200 μL). A drop of NH_4_OH solution was added to the solution followed by 2% sodium borohydride solution in dimethylsulfoxide (DMSO, 1.0 mL). The mixture was heated at 40 °C for 1 h and cooled down to room temperature. Glacial acetic acid (100 mL) was added followed by 1-methylimidazole (100 μL) and acetic anhydride (2.0 mL). The resulting reaction mixture was vortexed for 10 s and kept at room temperature for 10 min. The reaction was quenched by the addition of water (5.0 mL), extracted with CH_2_Cl_2_ (1.0 mL), and subjected to GC analysis with an Agilent Technologies 6890N GC System with 5975 Inert XL Mass Selective Detector using Supelco SP-2340 Capillary column (60 m × 0.25 mm × 0.25 μm film thickness). Standard sugars were used for identification.

### 3.10. Statistical Analysis

Statistical analyses, especially standard deviation, were calculated using Microsoft Excel Version 2021 (Microsoft Corp., Redmond, WA, USA).

## 4. Conclusions

*C. sorokiniana,* isolated from a pond adjacent to a cement plant, was cultured using flue gas in 20 L and 25,000 L photobioreactors. The resulting biomasses were freeze-dried and studied for their lipid, protein, and polysaccharide composition for possible application. The lipid content was observed to be slightly higher in biomass obtained from the 20 L reactors (21.54%) as compared to the 25,000 L (17.97%) photobioreactor. High concentrations of chlorophyll a and lutein were detected in both biomasses. Carotenoid concentration was in the range of 8.4–9.2 mg/g, and lutein accounts for 55% of the total carotenoids. The lipid was portioned into neutral lipid, glycolipid, and phospholipid fractions. UHPLC/HRMS analysis led to the identification of 71 intact TAGs from the neutral lipid fraction, as well as 8 LPCs and 10 PCs from the phospholipid fraction. Galactolipids, especially MGDGs, DGDGs, and a SQDG, were identified in both glycolipid and phospholipid fractions. This is the first report of the characterization of intact triacylglycerols in *C. sorokiniana*. To the best of our knowledge, this is the first comprehensive analysis of the lipids from *C. sorokiniana* characterizing TAGs, LPCs, PCs, SQDG, and DGDGs. The residual biomass after lipid extraction was further studied to characterize proteins and polysaccharides. Based on the amino acid analysis, the purity of protein isolates (PIs) was determined to be equal to or higher than 75%, with all essential amino acids present. Aspartic acid, glutamine, glycine, arginine, alanine, valine, isoleucine, and phenylalanine were identified as major amino acids. Monomer analysis of polysaccharide suggested that it was composed mainly of D-(+)-mannose, D-(+)-galactose and D-(+)-glucose. Comparing the current findings with the literature, the results demonstrate that there is no adverse effect on the metabolite profile of *C. sorokiniana* biomass cultured using flue gas as the primary carbon source. This opens the possibility for utilizing such algal biomass in industrial applications including food, feed, and biofuel; however, there is a need to monitor for the presence of potentially toxic chemicals in the algal biomass, such as heavy metals which may be present in some industrial flue gases.

## Figures and Tables

**Figure 1 molecules-29-03368-f001:**
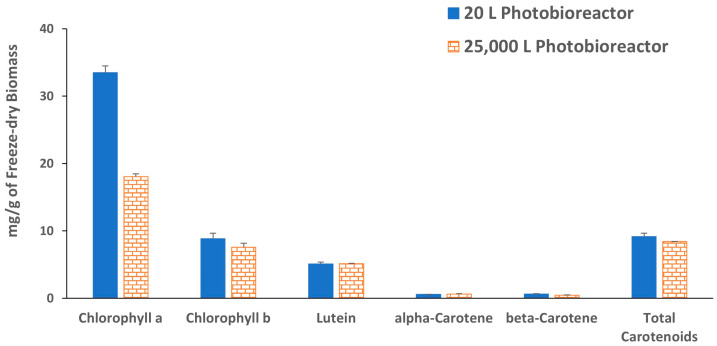
Concentration of chlorophyll a, chlorophyll b and carotenoids in *C. sorokiniana* freeze-dry biomasses. Each value represents the mean ± standard deviation (SD) of three independent experiments.

**Figure 2 molecules-29-03368-f002:**
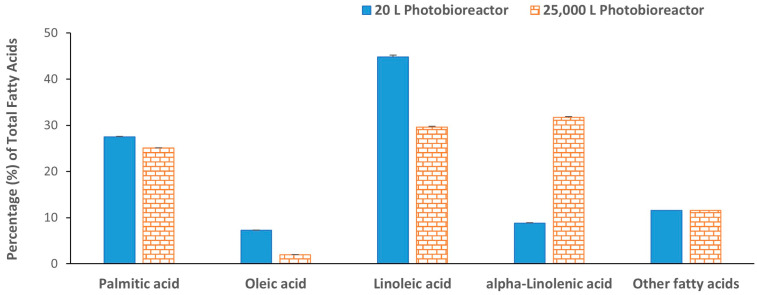
Percentage fatty acid distribution in lipid extracted from *C. sorokiniana* biomasses. Each value represents the mean ± SD of three independent experiments.

**Figure 3 molecules-29-03368-f003:**
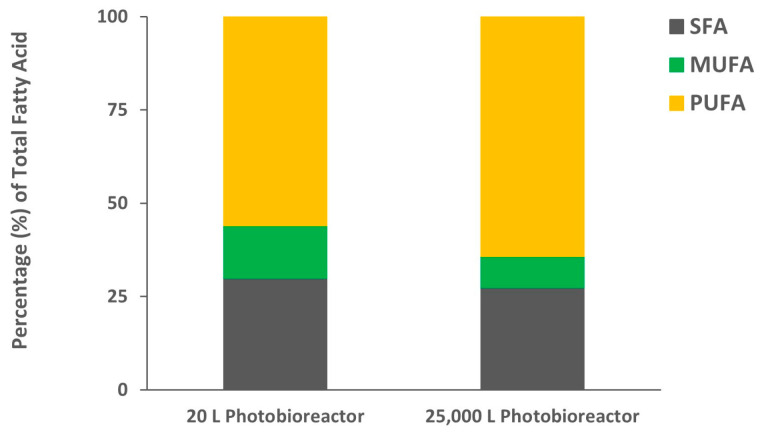
Percentage distribution of saturated fatty acid (SFA), monounsaturated fatty acid (MUFA), and polyunsaturated fatty acid (PUFA) in lipids extracted from *C. sorokiniana* biomass.

**Figure 4 molecules-29-03368-f004:**
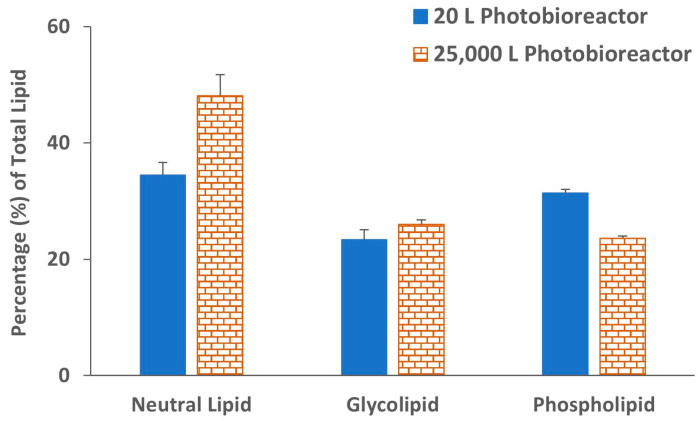
Percentage of neutral lipid, glycolipid, and phospholipid present in total lipid determined by silica gel-based solid phase extraction (SPE) of *C. sorokiniana* biomasses. Each value represents the mean ± SD of three independent experiments.

**Figure 5 molecules-29-03368-f005:**
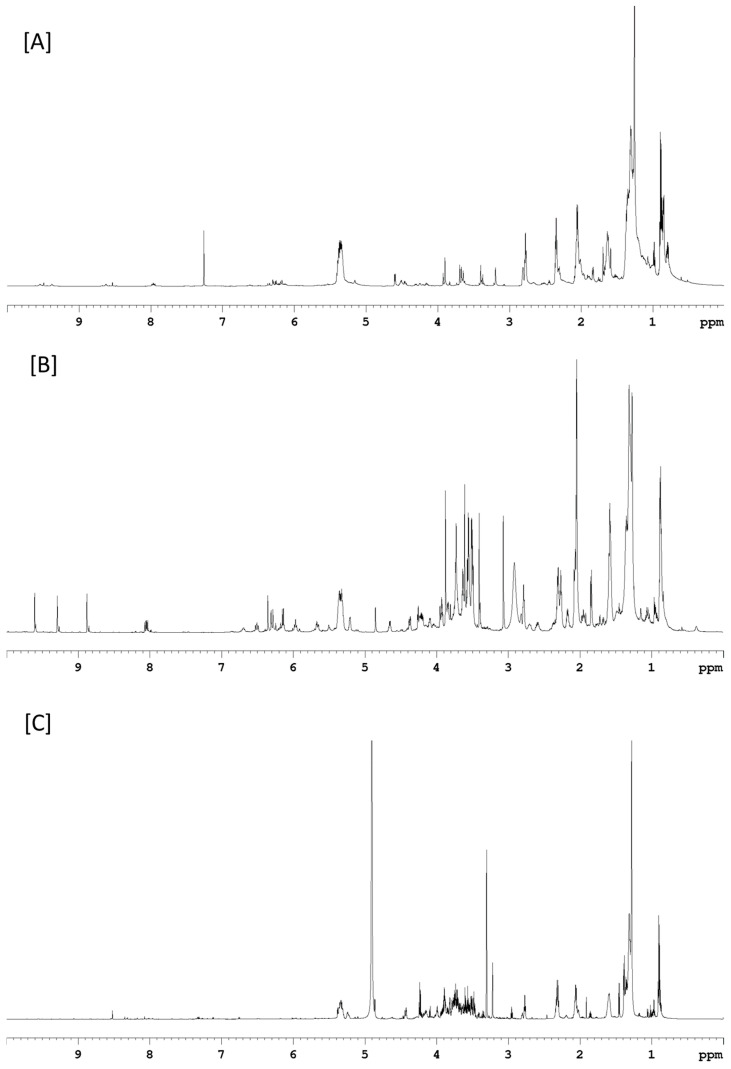
The ^1^H NMR spectra of (**A**) neutral lipid, (**B**) glycolipid, and (**C**) phospholipid fractions derived from lipid extracted from *C. sorokiniana* biomass cultured in the 20 L photobioreactor.

**Figure 6 molecules-29-03368-f006:**
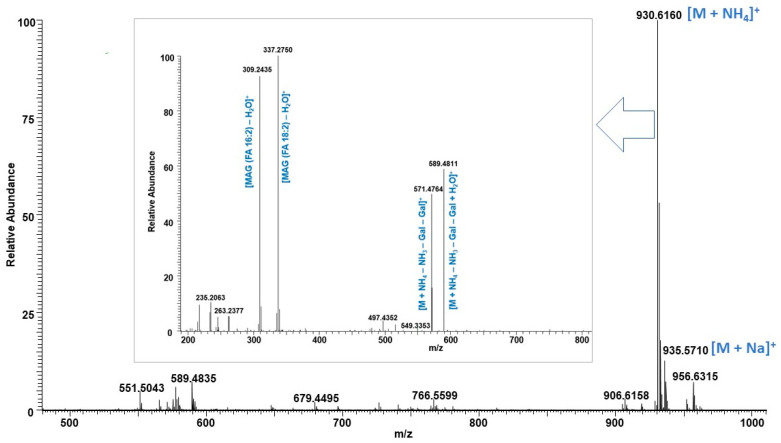
Mass spectrum of DGDG (34:4) ammonium adduct ion at *m*/*z* 930.6160 [M + NH_4_]^+^ with fragmentation ions observed in MS/MS spectrum shown in inset.

**Figure 7 molecules-29-03368-f007:**
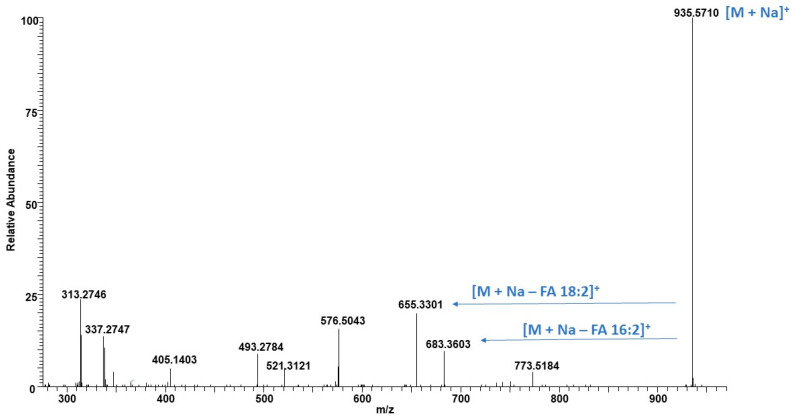
MS/MS spectrum of DGDG (34:4) sodium adduct ion at *m*/*z* 935.5710 [M + Na]^+^ with fragmentation ions.

**Figure 8 molecules-29-03368-f008:**
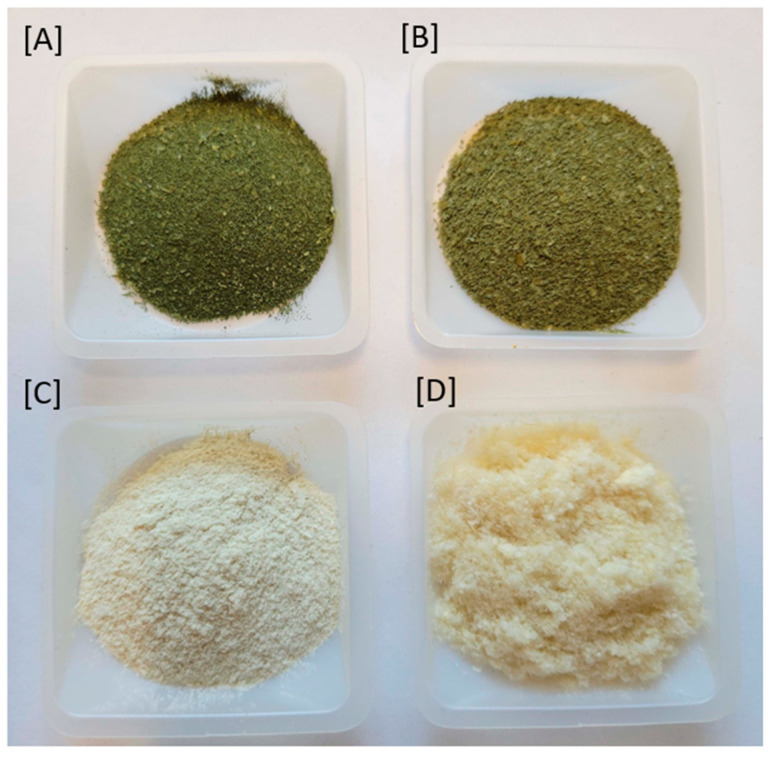
Protein isolates (PIs) extracted from *C. sorokiniana* cultured at 25,000 L (**A**) and 20 L (**B**) photobioreactors, and polysaccharide extracted from *C. sorokiniana* cultured at 25,000 L (**C**) and 20 L (**D**) photobioreactors.

**Figure 9 molecules-29-03368-f009:**
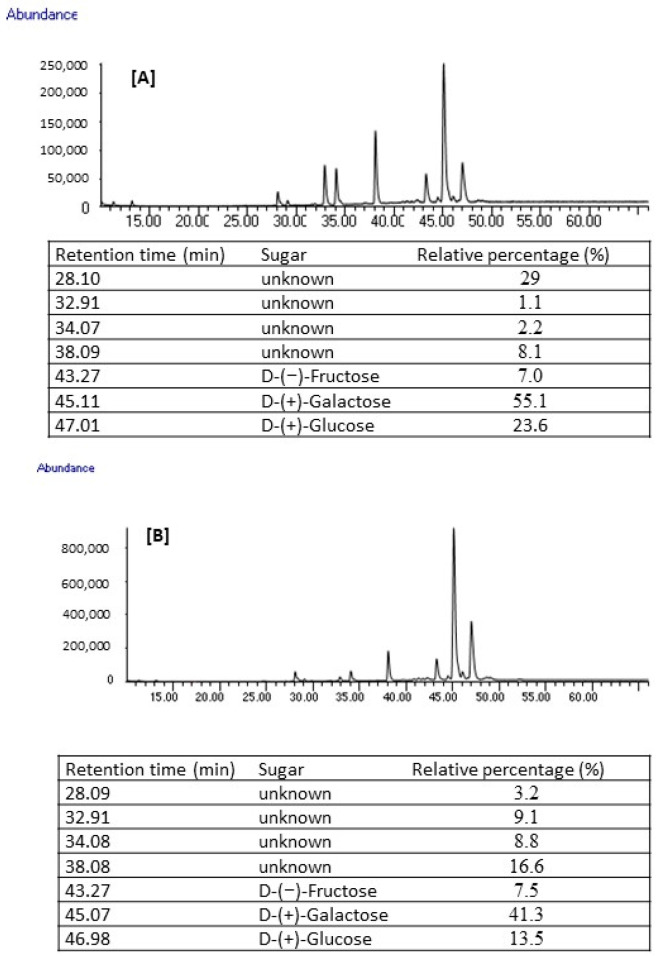
Sugar profile of polysaccharide extracted from (**A**) 20 L photobioreactor, (**B**) 25,000 L photobioreactor.

**Table 1 molecules-29-03368-t001:** Triacylglycerols (TAGs) detected in the neutral lipid fraction as ammonium adduct ion (M + NH_4_)^+^ and their relative abundance.

RT (min)	Measured ms (*m*/*z*)	Exact ms (*m*/*z*)	Error ppm *	TAG C:DB	TAG Identity	RA 20 L	RA 25,000 L
1.55	838.6931	838.6919	−1.4	50:7	18:3/16:3/16:1; 18:3/18:2/16:2; 18:2/16:3/16:2	3.2	0.0
1.55	864.7107	864.7076	−3.6	52:8	18:3/18:1/16:4; 18:3/18:2/16:3	1.2	0.0
1.84	840.7073	840.7076	0.4	50:6	18:3/16:3/16:0; 18:3/16:2/16:1; 18:2/16:3/16:1; 18:2/16:4/16:0	7.0	6.9
1.84	866.7231	866.7232	0.1	52:7	18:3/18:1/16:3; 18:2/18:1/16:4	4.1	8.1
1.95	894.7540	894.7545	0.6	54:7	18:3/18:2/18:3; 18:3/18:3/18:1	1.2	3.0
2.10	816.7070	816.7076	0.7	48:4	16:4/16:0/16:0; 16:3:16:1/16:0; 16:2/16:2/16:0	4.0	1.5
2.10	842.7226	842.7232	0.7	50:5	18:3/16:2/16:0; 18:3/16:1/16:1; 18:2/16:3/16:0; 18:2/16:2/16:1; 18:1/16:3/16:2; 18:1/16:4/16:0	8.0	7.5
2.10	868.7383	868.7389	0.7	52:6	18:3/18:2/16:1; 18:3/18:3/16:0; 18:1/18:0/16:5; 18:3/18:1/16:2; 18:1/18:1/16:4; 18:2/18:1/16:3	5.5	7.7
2.35	818.7233	818.7232	−0.1	48:3	16:3/16:0/16:0; 16:2/16:1/16:0	2.7	1.9
2.35	844.7388	844.7389	0.1	50:4	18:1/16:3/16:0; 18:1/16:2/16:1; 18:2/16:2/16:0; 18:2/16:1/16:1; 18:3/16:1/16:0	7.1	5.4
2.35	870.7543	870.7545	0.2	52:5	18:3/18:2/16:0; 18:2/18:2/16:1; 18:3/18:1/16:1; 18:2/18:1/16:2; 18:1/18:1/16:3; 18:0/16:2/16:3	4.4	7.8
2.35	896.7702	896.7702	0.0	54:6	18:3/18:2/18:1	1.4	3.2
2.60	820.7392	820.7389	−0.4	48:2	16:2/16:0/16:0; 16:1/16:1/16:0; 18:2/16:0/14:0; 18:1/16:0/14:1; 18:1/16:1/14:0	5.3	1.8
2.85	898.7859	898.7858	−0.1	54:5	18:3/18:1/18:1; 18:3/18:2/18:0; 18:2/18:2/18:1	1.4	2.9
2.85	846.7547	846.7545	−0.2	50:3	18:3/16:0/16:0; 18:2/16:1/16:0; 18:1/16:1/16:1; 18:0/16:2/16:1	6.5	4.4
2.85	872.7702	872.7702	0.0	52:4	18:3/18:1/16:0; 18:2/18:2/16:0 (major); 18:3/18:0/16:0	9.6	9.2
3.10	900.8013	900.8015	0.2	54:4	18:2/18:2/18:0 (major); 18:3/18:1/18:0	1.2	2.7
3.10	874.7855	874.7858	0.3	52:3	18:2/18:1/16:0; 18:2/18:0/16:1	4.6	6.9
3.10	848.7704	848.7702	−0.2	50:2	18:2/16:0/16:0 (major); 18:1/16:1/16:0	12.3	6.0
3.10	862.7859	862.7858	−0.1	51:2	18:1/17:1:16:0; 18:2/17:0:16:0;	3.0	1.5
3.10	888.8016	888.8015	−0.1	53:3	18:2:/18:1/17:0	0.7	0.9
3.70	850.7860	850.7858	−0.2	50:1	18:1/16:0/16:0	1.9	2.8
3.70	876.8016	876.8015	−0.1	52:2	18:1/18:1/16:0	2.7	4.5
3.70	902.8174	902.8171	−0.3	54:3	18:1/18:1/18:1 (major); 18:2/18:1/18:0 (minor)	0.5	2.0
3.90	878.8168	878.8171	0.3	52:1	18:1/18:0/16:0	0.5	1.5

RA—Relative abundance; * error ppm = (exact mass − measured mass)/exact mass × 10^6^; low abundance 
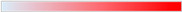
 high abundance.

**Table 2 molecules-29-03368-t002:** Free fatty acids and polar lipids identified in glycolipid fraction of *C. sorokiniana*. Proton adduct ion (M + H)^+^ was observed for fatty acids, and ammonium adduct ion (M + NH_4_)^+^ was observed for MGDGs, DGDGs, and SQDG.

RT (min)	Measured ms (*m*/*z*)	Exact ms (*m*/*z*)	Error ppm *	Lipid Identity	Fatty Acid Position R1/R2	RA 20 L	RA 25,000 L
Free Fatty Acids							
1.25	249.1850	249.1849	−0.4	FA 16:4		ND	ND
1.55	251.2005	251.2006	0.4	FA 16:3		ND	ND
1.95	277.2162	277.2162	0.0	FA 18:4		ND	ND
2.40	279.2318	279.2319	0.4	FA 18:3		ND	ND
MGDGs and DGDGs							
5.83	738.5156	738.5151	−0.7	MGDG 32:5	16:2/16:3	0.9	6.9
5.99	740.5312	740.5307	−0.7	MGDG 32:4	16:2/16:2	3.4	6.2
6.07	766.5477	766.5464	−1.7	MGDG 34:5	18:3/16:2	4.8	13.3
6.23	768.5622	768.5620	−0.3	MGDG 34:4	18:2/16:2	15.1	12.5
6.38	770.5777	770.5777	0.0	MGDG 34:3	18:2/16:1 ^#^	3.0	3.1
6.51	772.5936	772.5933	−0.4	MGDG 34:2	18:2/16:0 ^#^	1.7	1.2
6.60	764.5437	764.5307	17.0	MGDG 34:6	18:3/16:3	0.0	0.0
6.68	774.6092	774.6090	−0.3	MGDG 34:1	NI	0.9	0.6
6.82	776.6246	776.6246	0.0	MGDG 34:0	18:0/16:0 ^#^	1.1	0.6
5.85	926.5837	926.5835	−0.2	DGDG 34:6	18:3/16:3 ^#^	1.2	11.6
	902.5841	902.5835	0.7	DGDG 32:4	16:2/16:2, 16:3/16:1 ^#^	1.2	1.3
5.94	928.5995	928.5992	−0.3	DGDG 34:5	18:3/16:2; 18:2/16:3	7.4	11.7
6.10	930.6147	930.6148	0.1	DGDG 34:4	18:2/16:2	34.4	18.9
6.10	932.6299	932.6305	0.6	DGDG 34:3	18:3/16:0; 18:2/16:1; 18:1/16:2 ^#^	7.9	3.5
6.26	958.6458	958.6461	0.3	DGDG 36:4	18:2/18:2	4.3	2.5
6.45	934.6463	934.6461	−0.2	DGDG 34:2	18:2/16:0	9.2	4.8
6.45	936.6618	936.6618	0.0	DGDG 34:1	18:1/16:0 ^#^	3.6	1.4
SQDG							
5.94	812.5552	812.5552		SQDG 32:0	16:0/16:0		

RA—Relative abundance; ND—not determined; NI—not identified; * error ppm = (exact mass − measured mass)/exact mass × 10^6^; ^#^ position of acyl chain is not assigned because of poor fragmentation of [M + Na]^+^ adduct ion or presence of multiple compounds with identical mass; low abundance 
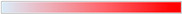
 high abundance.

**Table 3 molecules-29-03368-t003:** Polar lipids identified in phospholipid fractions of *C. sorokiniana.* Proton adduct ion (M + H)^+^ was observed for LPCs and PCs, and ammonium adduct ion (M + NH_4_)^+^ was observed for MGDG, DGDGs, and SQDG.

RT (min)	Observed Mass (*m*/*z*)	Exact Mass (*m*/*z*)	Error ppm *	Lipid Identity	Fatty Acid Position R1/R2	RA 20 L	RA 25,000 L
LPCs and PCs							
2.55	490.2929	490.2928	−0.2	LPC 16:3	16:3/0:0	0.1	3.4
3.10	492.3086	492.3085	−0.2	LPC 16:2	16:2/0:0	1.0	2.9
3.10	494.3252	494.3241	−2.2	LPC 16:1	16:1/0:0	0.2	0.9
3.45	518.3244	518.3241	−0.6	LPC 18:3	18:3/0:0	0.4	9.0
3.96	520.3402	520.3398	−0.8	LPC 18:2	18:2/0:0	6.2	7.3
4.25	496.3400	496.3398	−0.4	LPC 16:0	16:0/0:0	1.9	6.0
4.48	522.3582	522.3554	−5.4	LPC 18:1	18:1/0:0	0.0	0.6
4.94	524.3742	524.3711	−5.9	LPC 18:0	18:0/0:0	0.1	0.0
5.30	766.5019	766.5017	−0.3	PC 34:6; O	NI	0.0	3.8
5.50	768.5175	768.5174	−0.1	PC 34:5;O	NI	0.8	3.5
5.50	794.5332	794.5330	−0.3	PC 36:6; O	NI	0.3	2.7
6.40	780.5535	780.5538	0.4	PC 36:5	NI	9.3	8.4
5.50	754.5379	754.5381	0.3	PC 34:4	NI	4.1	8.6
5.50	728.5227	728.5225	−0.3	PC 32:3	NI	7.0	9.9
6.60	782.5695	782.5694	−0.1	PC 36:4	NI	11.4	10.8
6.60	756.5539	756.5538	−0.1	PC 34:3	NI	12.0	8.3
6.95	784.5853	784.5851	−0.3	PC 36:3	NI	41.1	11.3
6.80	758.5696	758.5694	−0.3	PC 34:2	NI	4.1	2.7
MGDG and DGDGs							
5.92	764.5437	764.5307	17.0	MGDG 34:6	18:3/16:3	5.1	29.8
5.92	926.5840	926.5835	−0.5	DGDG 34:6	18:3/16:3	1.2	12.3
5.94	902.5841	902.5835	−0.7	DGDG 32:4	16:2/16:2; 16:3/16:1 ^#^	1.7	1.7
5.94	928.5988	928.5992	0.4	DGDG 34:5	18:3/16:2; 18:2/16:3 ^#^	11.8	22.2
6.14	930.6156	930.6148	−0.9	DGDG 34:4	18:2/16:2	49.5	20.0
6.24	932.6296	932.6305	1.0	DGDG 34:3	18:3/16:0; 18:2/16:1; 18:1/16:2 ^#^	8.0	3.8
6.25	958.6458	958.6461	0.3	DGDG 36:4	18:2/18:2	4.4	2.6
6.38	934.6461	934.6461	0.0	DGDG 34:2	18:2/16:0	12.9	6.1
6.50	936.6614	936.6618	0.4	DGDG 34:1	18:1/16:0	5.4	1.6
SQDG							
5.94	812.5552	812.5552	0.0	SQDG 32:0	16:0/16:0		

RA—Relative abundance; NI—not identified; * error ppm = (exact mass − measured mass)/exact mass × 10^6^; ^#^ position of acyl chain are not assigned because of poor fragmentation of [M + Na]^+^ adduct ion or presence of multiple compounds with identical mass; low abundance 
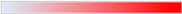
 high abundance.

**Table 4 molecules-29-03368-t004:** Amino acid analysis of crude protein isolates (PIs) extracted from *C. sorokiniana* cultured with the flue gas (mg/g freeze-dried biomass). Each value represents the mean ± SD of three independent experiments.

Amino Acids (AA)	20 L Photobioreactor	25,000 L Photobioreactor
Aspartic acid (Asp)	59.6 ± 2.1	66.4 ± 3.1
Serine (Ser)	25.0 ± 0.2	34.9 ± 1.7
Glutamic acid (Glu)	75.4 ± 1.9	84.8 ± 4.4
Glycine (Gly)	50.7 ± 1.3	59.1 ± 2.7
Histidine (His)	22.9 ± 0.6	26.2 ± 1.4
Arginine (Arg)	56.1 ± 2.6	66.9 ± 2.8
Threonine (Thr)	24.7 ± 0.5	33.8 ± 1.6
Alanine (Ala)	48.0 ± 0.5	53.9 ± 2.7
Proline (Pro)	37.2 ± 0.4	45.3 ± 2.2
Cysteine (Cys)	-	-
Tyrosine (Tyr)	48.0 ± 2.1	57.6 ± 2.3
Valine (Val)	52.1 ± 1.5	59.3 ± 3.5
Methionine (Met)	21.2 ± 2.3	22.5 ± 3.0
Lysine (Lys)	34.9 ± 2.2	41.6 ± 2.6
Isoleucine (Ile)	37.5 ± 1.2	39.7 ± 2.4
Leucine (Leu)	86.9 ± 1.3	89.1 ± 4.3
Phenylalanine (Phe)	68.7 ± 2.6	71 ± 3.8
Total amino acids (mg/g)	748.9 ± 6.4	852.2 ± 39.1

(-) not detected.

## Data Availability

The original contributions presented in the study are included in the article/Appendix A, further inquiries can be directed to the corresponding author/s.

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
