# Peer review of "Comprehensive Analysis of Biomass from Chlorella sorokiniana Cultivated with Industrial Flue Gas as the Carbon Source"

_molecules, 2024, doi:10.3390/molecules29143368_

Round 1

Reviewer 1 Report

Comments and Suggestions for Authors

The present study titled " Comprehensive Analysis of Biomass from Chlorella sorokiniana Cultivated with Industrial Flue Gas as the Carbon Source" by Banskota et. al., is particularly interesting. The quality of the writing is good. There are a few comments to improve the manuscript:

- Please use formal language in the text without using "we

-The quality of all figures should be increased. Especially, NMR spectrums (Figure 5).

-Please reorganize the abstract section to include brief information about the article's topic, objectives, the experiments conducted in the study, and the results obtained.

- All abbreviations and words that need to be italicized in the text should be checked

-What is cement kiln? Please give definition in the text

-Summarized legend will be helpful to understand the figures

-Please include method of statistical analysis. Which software and statistical approaches did you used?

-Please also indicate novelty of your research article

- The conclusion section should be added to highlight future remarks, importance, and relevance of their work.

Author Response

We appreciate your comments and positive feedback. The following changes have been made to the revised manuscript:

1:  Please use formal language in the text without using "we”

Response: Instead of "we" or "our," the publication is now cited in lines 70, 100 and 261.

2:  The quality of all figures should be increased. Especially, NMR spectrums (Figure 5)

Response: Figure 5 has been updated to enhance the quality with better resolution.

3:  Please reorganize the abstract section to include brief information about the article's topic, objectives, the experiments conducted in the study, and the results obtained

Response: The abstract has been revised and the total word count is now under 200 words, in accordance with the journal's guidelines.

4:  All abbreviations and words that need to be italicized in the text should be checked

Response: The scientific names of microalgae have all been italicized throughout the manuscript, and the abbreviations have been properly introduced.

5:  What is cement kiln? Please give definition in the text

Response: The phrase “cement kiln” is replaced with " kiln" in the abstract and text.

6:  Summarized legend will be helpful to understand the figures

Response: The legends of the figures and the titles of the tables have been revised to summarize the results.

7:  Please include method of statistical analysis. Which software and statistical approaches did you used?

Response: All statistical analyses were performed using Microsoft Excel Version 2021 and new statement is added in the experimental section 3.9.

8:  Please also indicate novelty of your research article

Response: A new statement is added in the discussion to highlight the novelty of the work.

9:  The conclusion section should be added to highlight future remarks, importance, and relevance of their work

Response: Conclusion is revised.

Reviewer 2 Report

Comments and Suggestions for Authors

This work has investigated the comprehensive analysis of biomass from Chlorella sorokiniana cultivated with industrial flue gas as the carbon source. The results have been proved to be essential for this paper and appropriate to be published. However, the authors should address the following issues and pass through a major revision for improvement. A list of comments is provided below.

1. Please revise the Abstract to 200 words maximum.

2. In the Introduction, the issue needs to be well clarified why Chlorella sorokiniana was chosen as the targeting bioresource compared to others.

3. In the Introduction, please refer the previous studies to current research in the past 5-8 years.

4. Please make 1 space between digits and L in the whole manuscript (for example, 20 L and 25000 L).

5. The abbreviations 1H NMR, HPLC, LC/MS, and GC should be provided with their full names when first introduced in the Introduction. However, the other mentioned can be written in the form of the abbreviation.

6. The order of manuscript should be 1. Introduction, 2. Materials and Methods, 3.Results and Discussion, 4. Conclusions, please revise the manuscript.

7.Please perform statistical tests for comparison of your results. Also provide standard deviation of each mean value. The results of the statistical tests must be shown in the manuscript, please revise the manuscript.

8. In the section of Moisture, ash, lipid content, and lipid class separation and Fatty Acid Analysis, please separate formula from the text and provide (1), (2), (3), and (4).

9. Please specify the wording "minor modification" and "slight modifications".

10. Keywords should be ordered in alphabetical order.

11. Please recalculate rpm on g.

12. Why compare algal biomasses obtained from 20 L and 25000 L Photobioreactors?

13. Please revise the conclusion and mention only key findings.

14. What are the key motivations for using industrial flue gas as a carbon source for cultivating Chlorella sorokiniana?

15. How does the composition of industrial flue gas affect the growth rate and biomass yield of Chlorella sorokiniana?

16. How does the quality of biomass produced using industrial flue gas compare to that produced using conventional carbon sources?

Author Response

We highly appreciate the thorough review of our manuscript. The following changes have been made to the revised manuscript, and we are hopeful that these will address all the questions and suggestions raised.

1: Please revise the Abstract to 200 words maximum.

Response: The abstract has been revised and the total word count is now under 200 words, in accordance with the journal's guidelines.

2: In the Introduction, the issue needs to be well clarified why Chlorella sorokiniana was chosen as the targeting bioresource compared to others.

Response: A new statement has been added to the introduction section to clarify why Chlorella sorokiniana was chosen for further biomass production.

3: In the Introduction, please refer the previous studies to current research in the past 5-8 years.

Response: An additional paragraph has been added to the introduction section, describing the latest research on carbon dioxide fixation using microalgae and flue gas as a carbon source.

4: Please make 1 space between digits and L in the whole manuscript (for example, 20 L and 25000 L).

Response: Space is added between the digit and "L" (i.e., 20L to 20 L) throughout the manuscript and in Figures/Tables.

5: The abbreviations 1H NMR, HPLC, LC/MS, and GC should be provided with their full names when first introduced in the Introduction. However, the other mentioned can be written in the form of the abbreviation.

Response: The full forms of 1H NMR, HPLC, LC/MS, and GC are provided in the text when first used, followed by their abbreviations in the rest of the manuscript.

6: The order of manuscript should be 1. Introduction, 2. Materials and Methods, 3. Results and Discussion, 4. Conclusions, please revise the manuscript.

Response: The manuscript follows the Journal's guidelines, structured with sections for introduction, results, discussion, materials and methods, and conclusion. If the Editor prefers, we would be happy to rearrange the order as per the reviewer’s suggestions.

7: Please perform statistical tests for comparison of your results. Also provide standard deviation of each mean value. The results of the statistical tests must be shown in the manuscript, please revise the manuscript.

Response: All statistical analyses were performed using Microsoft Excel Version 2021 and added in the experimental section 3.9.

8: In the section of Moisture, ash, lipid content, and lipid class separation and Fatty Acid Analysis, please separate formula from the text and provide (1), (2), (3), and (4).

Response: The formulas were separated from the text and listed as (A), (B), (C), and (D) in section 3.3 and 3.5.

9: Please specify the wording "minor modification" and "slight modifications".

Response: To simplify, the words "minor" and "slight" were removed from the experimental section, and modifications were stated directly. The actual experimental protocol was provided briefly in the experimental section.

10: Keywords should be ordered in alphabetical order.

Response: Keywords are rearranged in alphabetical order.

11: Please recalculate rpm on g.

Response: RPM has been changed to g.

12: Why compare algal biomasses obtained from 20 L and 25000 L Photobioreactors?

Response: The algal biomasses obtained from the two photobioreactor scales, 20 L and 25000 L, were studied in order to understand if there is any difference their composition. The two photobioreactors have some difference in size and light delivery.

13: Please revise the conclusion and mention only key findings.

Response: Conclusion is revised.

14: What are the key motivations for using industrial flue gas as a carbon source for cultivating Chlorella sorokiniana?

Response: Brief statements are added in both the introduction and Results and Discussion sections describing the objective of the study with reference to our previous publication describing 28 microalgae strains collected in the vicinity of a cement production plant and the screening of these strains in raw kiln gas from the same plant.  The C. sorokiniana strain (SMC-14) that showed the most tolerance to cement kiln gas emissions in situ was the strain used to produce the biomass analyzed in our current study.  In addition, in the introduction section of our current manuscript we describe a number of the attributes of genus Chlorella, and C. sorokiniana in particular, that make it attractive for commercial production.

15: How does the composition of industrial flue gas affect the growth rate and biomass yield of Chlorella sorokiniana?

Response: It is expected that an increase of CO2 in the incoming gas will improve the growth rate in a healthy, well illuminated microalge culture until the point where the growth medium becomes excessively acidified for the cultivated strain.  Other gasses present in the kiln gas, including CO, SOX and NOX are expected to become inhibitory to growth when they are present in concentrations above a species-specific threshold.  We did not test C. sorokiniana SMC-14's tolerance to a range of concentration of CO2 or the other constituent gasses from cement kiln emissions in this this study. We have reported growth rate and biomass production of C. sorokiniana (SMC 14M) with flue gas in our earlier report and a statement with growth rate and biomass yield are added in Results and Discussion section of the current manuscript with the appropriate reference.

16: How does the quality of biomass produced using industrial flue gas compare to that produced using conventional carbon sources?

Response: While comparing the composition of the algal biomass with or without flue gas is interesting, no biomass was generated using conventional carbon sources. Our R&D project was focused on growing microalgae in flue gas and scaling up experiments.

Round 2

Reviewer 2 Report

Comments and Suggestions for Authors

In the revised manuscript, the authors have made sufficient modifications according to the modification comments. Therefore, after revision, the work of this manuscript is more clear. However, this revised manuscript still has the following problems:

1. The authors said "The high-value products such as pigments including carotenoids, both polar and non-polar lipids were characterized by NMR, HPLC, LC/MS and GC analyses." in the Introduction. However, the abbreviations 1H NMR, HPLC, LC/MS, and GC should be provided with their full names when first introduced in the Introduction. It should be "proton nuclear magnetic resonance (1H NMR) NMR, High Performance Liquid Chromatography (HPLC), liquid chromatography/mass spectrometry (LC/MS) and Gas chromatography (GC) analyses.".

2. Please provide the method for mean comparison in the experimental section 3.9 in detail. For example, "All statistical analyses were performed using the Statgraphics Plus program, version 5.1 (Manugistics Inc., Rockville, MD, USA). The means were compared using Fisher's least significant difference (LSD) test and the statistical significance was determined at P < 0.05."

3.Base on the Journal's guidelines, the formulas should be separated from the text and listed as (1), (2), (3), and (4) in section 3.3 and 3.5. Please check and revise the manuscript.

4. In section 3.7. Protein isolate (PI) and polysaccharide extraction, please recalculate 4000 rpm on g.

5. The algal biomasses obtained from the two photobioreactor scales, 20 L and 25000 L, were studied in order to understand if there is any difference in their composition. The two photobioreactors have some difference in size and light delivery. However, there is a very big gap between 20 L and 25000 L. Why not 100 L, 1000 L and 10,000 L Photobioreactors?

Author Response

We appreciate your comments and positive feedback. The following changes have been made to the revised manuscript:

  1. The authors said "The high-value products such as pigments including carotenoids, both polar and non-polar lipids were characterized by NMR, HPLC, LC/MS and GC analyses." in the Introduction. However, the abbreviations 1H NMR, HPLC, LC/MS, and GC should be provided with their full names when first introduced in the Introduction. It should be "proton nuclear magnetic resonance (1H NMR) NMR, High Performance Liquid Chromatography (HPLC), liquid chromatography/mass spectrometry (LC/MS) and Gas chromatography (GC) analyses."

Response: The full forms of 1H NMR, HPLC, LC/MS, and GC are provided in the introduction.

  1. Please provide the method for mean comparison in the experimental section 3.9 in detail. For example, "All statistical analyses were performed using the Statgraphics Plus program, version 5.1 (Manugistics Inc., Rockville, MD, USA). The means were compared using Fisher's least significant difference (LSD) test and the statistical significance was determined at P < 0.05."

Response: The company name and software used for calculating standard deviation is added in section 3.9. Moreover, additional statement is provided in the legend of figures and in section 3.5 describing the total number of experiments.

  1. Base on the Journal's guidelines, the formulas should be separated from the text and listed as (1), (2), (3), and (4) in section 3.3 and 3.5. Please check and revise the manuscript.

Response: The formulas are separated from the text in sections 3.3 and 3.5, and are listed as (1), (2), (3) and (4).

  1. In section 3.7. Protein isolate (PI) and polysaccharide extraction, please recalculate 4000 rpm on g.

Response: RPM is changed into g in section 3.7.

  1. The algal biomasses obtained from the two photobioreactor scales, 20 L and 25000 L, were studied in order to understand if there is any difference in their composition. The two photobioreactors have some difference in size and light delivery. However, there is a very big gap between 20 L and 25000 L. Why not 100 L, 1000 L and 10,000 L Photobioreactors?

Response: Although C. sorokiniana was cultured at various sizes (20 L, 1000 L, and 25000 L) photobioreactors, a comparison between the 20 L photobioreactors and 25000 L photobioreactor was made because both systems used the same monochromatic LED (640 nm wavelength) light source, while the 1000 L photobioreactors used fluorescent lights, invalidating its comparison to the other two.